# Using LSTM and PSO techniques for predicting moisture content of poplar fibers by Impulse-cyclone Drying

**Feng Chen** [1]*, **Xun Gao**[2], **Xinghua Xia**[1], **Jing Xu**[3]

**1** School of Art and Design, Taizhou University, Taizhou, Zhejiang, China, **2** College of Civil Engineering, Hunan University, Changsha, Hunan, China, **3** College of Material Science and Engineering, Northeast Forestry University, Harbin, Heilongjiang, China

☉ These authors contributed equally to this work.
* chenfeng1984@tzc.edu.cn

**Data Availability Statement:** All relevant data are within the manuscript and its Supporting Information files.

**Funding:** This research was funded by the National Natural Science Foundation of China (Grant

## Abstract

Impulse-cyclone drying (ICD) is a new type of pretreatment method to remove the excess moisture of wood fibers (WFs) with high speed and low energy consumption. However, the process parameters are often determined by the experience of the process operators, thus the quality of WF drying lacks an objective basis and cannot be ensured. To address this issue, this study adopted the long short-term memory (LSTM) neural network, backpropagation neural network, and Central-Composite response surface method to establish a moisture content (MC) prediction model and a process parameter optimization model based on single-factor experiments. The initial MC, inlet air temperature, feed rate, and inlet air velocity were taken as the experimental factors, and the final MC was taken as the inspection index. The parameters of LSTM were optimized by particle swarm optimization (PSO) algorithm, and the predicted value of MC was fitted to the model. The PSO-optimized LSTM had higher prediction accuracy than did the typical prediction models. The optimal process for the targeted MC, which was obtained by PSO, was featured with an initial MC of 10.3%, inlet air temperature of 242˚C, feed rate of 90 kg/h, and inlet air velocity of 8 m/s. PSO-LSTM could be a new approach for predicting the MC of WFs, which, in turn, could provide a theoretical basis for the application of ICD technology in the biomass composite industry.

## Introduction

The moisture content (MC) in wood fibers (WFs) not only has an important influence on the strength and other mechanical properties of wood–plastic composites (WPCs) but also is an important factor affecting the deformation of WPCs [1]. The main chemical components of wood cell wall include hydroxyl groups with strong water absorption and other oxygen-containing groups that can form hydrogen bonds with water [2]. Because of this hygroscopicity, the properties of the composite processing and final products may be affected. WFs have particularly high MC, and thus they should be dried to obtain a certain MC before compounding [3]. In high-temperature composition, the moisture is rapidly vaporized from the fiber surface

No.31901243). The funders had no role in study design, data collection and analysis, decision to publish, or preparation of the manuscript.

**Competing interests:** The authors declare no conflict of interest.

and forms weak boundary layers at the interface of the composite, which may lead to micropores or internal stress defects in the composite and affect the material properties (Fig 1). The exposed fibers of the boundary layer may lead to water absorption and expansion [4]. The WPC may prevent the expansion pressure from producing pores or microcracks to release the stress, and thus its properties deteriorate. Some scholars have found that the optimum MC for WFs in WPCs is 1–3% [5, 6].

The drying process of WFs is an important pretreatment process of WPCs. The most common drying methods include oven-drying, drum-drying, pipeline air drying, impinging stream drying, and infrared drying [7–10]. However, these drying techniques require long drying time and high energy consumption to remove bound water from fibers; furthermore, they are complex to operate and expensive to maintain and thus difficult to be applied and popularized in the industry. To solve these problems, the studies of many industries, such as food, medicine, medium fiberboards, and chemical industries, have focused on the selection of drying process parameters and the development of multistage drying techniques that combine different types of dryers [11, 12]. However, there are few reports on the application of such techniques to WFs used in WPCs; therefore, this topic should be further studied. Impulse-cyclone drying (ICD) has a short drying time (8–10 s) and has high mass transfer efficiency [11]. The process parameters are often determined by the experience of the process operators, thus the quality of WF drying lacks an objective basis and cannot be ensured. It is difficult to obtain the targeted MC by directly controlling the inlet temperature, air velocity, and feed rate, which limits its drying advantages and hinders the popularization and application of this technology in WPCs production enterprises.

Therefore, the understanding of moisture drying mechanism and the prediction of MC in drying process have attracted much scholarly attention. Optimization is time-consuming and cumbersome because a large number of tests are required to obtain the desired results. This can be solved by appropriate mathematical models, when the correlation between various input and output variables is used for simulation [13, 14].

For the prediction of MC, the existing traditional prediction methods, such as linear regression and time series, are more mature [15]. However, the initialization of the above models is difficult, and the process is complex and requires extensive experience and special skills. The prediction accuracy of the MC of WF also requires high data integrity and diversity. The insufficient consideration of key factors such as temperature and fiber MC may greatly reduce the prediction effect of the model. Previous studies have proposed various models of artificial intelligence prediction methods based on statistics and deep learning, including feed-forward neural network, backpropagation (BP) neural network, fuzzy neural model, support vector machine (SVM) model, and a model that combines Particle Swarm Optimization (PSO) and SVM [16]. However, the prediction performance of models trained with different parameters

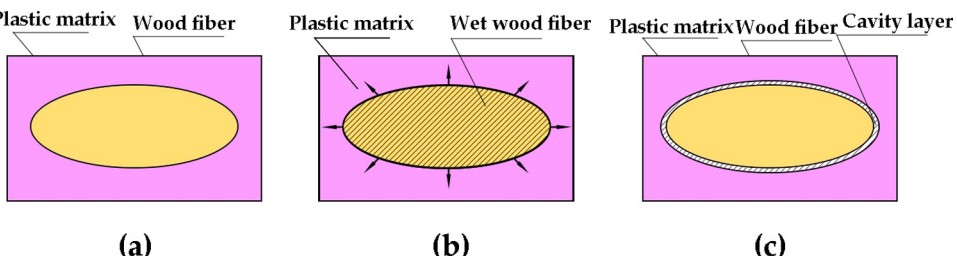

**Fig 1.** Damage principle of moisture in WFs to wood plastic composites, (a) dry WPC, (b) Moisture evaporation of WF, (c) Formation of cavity layer after evaporation.

varies greatly. The parameters set by experience may lead to instable prediction results and reduce the prediction accuracy. In addition, neural network models are prone to gradient disappearance or gradient explosion [17–20]. For the establishment of a MC prediction model for WF drying, there is no relevant literature. As an improved recurrent neural network (RNN), the long short-term memory (LSTM) neural network can effectively learn the long-term dependence of temporal data, and thus it is widely used in many scientific and technological fields, such as machine reading, emotion analysis, and image description [21, 22]. Moreover, the LSTM model is applied to the prediction of product performance under the interaction of multiple factors [23, 24]. The prediction effect of the LSTM model depends on the reliability of input variables, but there is no research exploring the combination of WF drying process conditions and LSTM. PSO was first proposed by Eberhart and Kennedy in 1995 [25, 26]. Its basic concept comes from the study of birds' foraging behavior. PSO algorithm is inspired by the behavior of this biological population and used to solve optimization problems [27]. Accordingly, PSO is used to optimize the LSTM to obtain the optimal parameters. Using the obtained hyperparameters to build the model and get the prediction results improves the prediction accuracy of the model greatly.

However, there is little research on the prediction of the MC. In this study, we analyzed the advantages of ICD drying over conventional oven-drying and drum-drying. To determine the relationship between the process parameters and the final MC, we used the initial MC, inlet airflow temperature, air velocity, and feed rate as input parameters and the final MC after drying as output parameter for the PSO-LSTM model. The traditional linear regression analysis, BP neural network model, and LSTM model were compared, and the model was validated by ICD experiments, which provided a theoretical basis for ICD process innovation and intelligent control. The results of this study could provide an effective method of MC prediction for the theoretical research of heat and mass transfer during drying and provide a reliable guarantee for the quality and efficiency of drying under ICD.

## Materials and methods

### Description of impulse-cyclone drying

The experiments were performed in an ICD system (MQG-50, Jianda Drying Equipment Co., Ltd., Changzhou, China), which is schematically presented in Fig 2. The ICD system was equipped with an electric heater, an induced draft fan, a screw feeder, an impulse dryer, a cyclone dryer, and a cyclone separator. The power absorbed by the induced draft fan could be modified within the range 0–66 000 W, and its speed could be adjusted from 1610 to 2844 $m^3 \cdot h^{-1}$. The poplar fibers could be put into the screw feeder with the flow regulation mode of frequency conversion speed regulation within the range 0–1100 W. The impulse dryer comprised of three pulse pipes (186 cm in height and 30 cm in diameter) and three straight pipes (18 cm in diameter) made of 0.2 cm thick steel sheet.

### Poplar fibers sample

Because poplar veneers are easier to obtain in a disintegrator with uniform aspect ratio and MC after milling, poplar veneers of about 500 kg (Zhonghan-17 fast-growing poplar, Harbin Yongxu Wood-Based Panel Co., Ltd., Heilongjiang, China) were selected as the fiber source. The size of each veneer was 1.2 mm × 40 mm × 40 mm, the air-dried density was 0.38 $g/m^3$, and the average MC was 13.8 (± 1.2) %. To satisfy the experimental requirements, the veneers were crushed into 60–80 mesh fiber samples by a biomass fiber crusher (MF-600, Jiangsu Fuyang Machinery Co., Ltd., Xuzhou, China), as shown in Fig 3. The material was passed through the screening machine to obtain the required form of fiber. By changing the screen

155

186

160

110

186

308 155

Wood fibers sample

**Dimensions in cm**

**Fig 2. Impulse-cyclone drying system used in the study.**

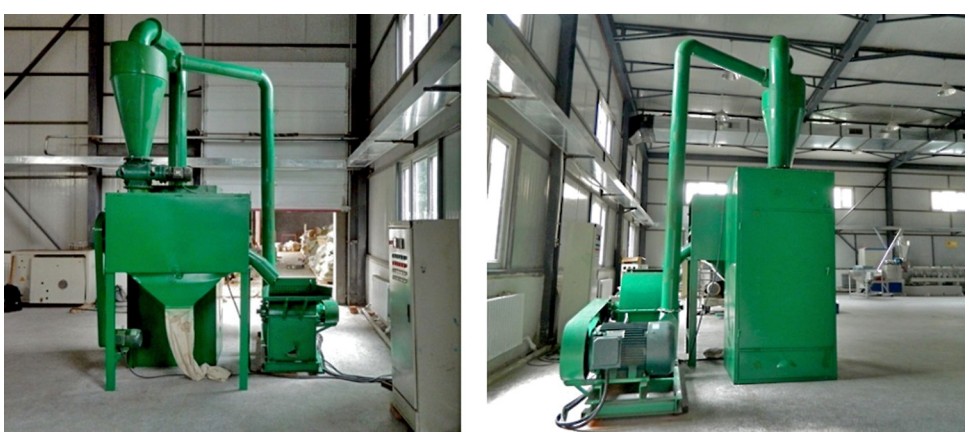

**Fig 3. Biomass fiber crusher.**

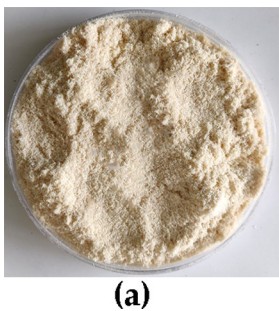 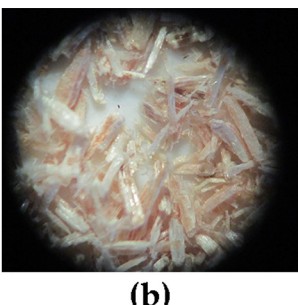 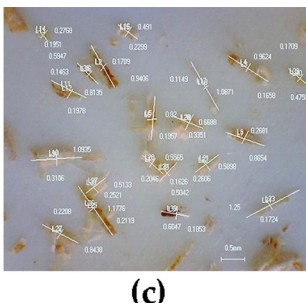

**(a)**                **(b)**                **(c)**

**Fig 4.** The measurement process of wood fiber morphology, (a) Poplar fiber sample, (b) Micromorphology of WFs under 80x eyepiece, (c) Measurement of aspect ratio under 40x HD digital microscope.

corresponding to the screening machine, 60–80 mesh WF could be obtained [28]. Then, the screened WFs were randomly sampled, and the samples were taken three times with 100 mg each time. Fig 4 shows the measurement process of WF morphology. The sample was placed on the glass slide and then analyzed by high-definition digital microscopes (GE-5, Shanghai Changfa Optical Instrument Co., Ltd., Shanghai, China) with a magnification of 40 times. The arithmetic mean value of the three sampling measurement results was calculated as the measurement result, with the average length of WFs, average diameter, and aspect ratio being 1.53 (± 0.18) mm, 283 (± 35) μm, and 5.4 (± 1.61), respectively.

## Drying experiments

In the experiment, the MC of WF was pretreated to obtain the initial MC. The high-pressure spray method was adopted to deal with the WF. The specific method was to put poplar fibers into the high-speed mixer and atomize them with high pressure sprayers at the feeding port to make the MC of poplar uniform. The fiber was extracted and sealed with a sealed bag and kept for 24 h, so that the MC was balanced. The WFs were first subjected to a drying environment generated using a heat generator, a screw feeder, and an induced draft fan under different temperatures (160–240°C), inlet air velocities (9–13 m/s), and feed rates (90–150 kg/h), with the hot air being the drying medium; the fibers were dried by heat and mass exchange with the hot airflow.

## Determination of moisture content

The average MC of poplar fiber was measured according to "Standard Test Methods for Direct Moisture Content Measurement of Wood and Wood-Based Materials" (ASTM D4442-2016). MC was calculated as follows:

$$MC, \% = (W_A - W_B)/W_B \times 100, \qquad (1)$$

where $W_A$ is the original mass (g), and $W_B$ is the ovendry mass (g).

## Different drying methods on MC uniformity and energy consumption

The fibers dried by different methods, such as ICD, oven-drying, and drum-drying, were randomly sampled, and 30 groups of samples were taken for MC determination. The weight of the fiber was 5 kg, the initial MC was 13±2%, and the fiber size was 60–80 mesh. The drying process conditions of ICD system were an inlet air temperature of 180°C, a feed rate of 90 kg/h, and an inlet air velocity of 9 m/s. The oven-drying conditions were a temperature of 103°C

and a drying time of 6 h. The process conditions of rotary drum-drying were a drum surface temperature of 120˚C, a rotating speed of 4 r/min, and a drying time of 15 min.

## Main structure of MC prediction model

The main structure of MC prediction model was shown in Fig 5.

## Multiple Linear Regression (MLR) model

According to the single-factor experimental study of ICD, combined with the actual operation results, the initial MC (A), inlet air temperature (B), feed rate (C), and inlet air velocity (D) were taken as the experimental factors, and the final MC was taken as the inspection index. To construct MLRs, the Central Composite Response Surface Methodology (RSM) design conditions were constructed using the statistical software package Design Expert 10.0.7, Stat-Ease Inc., MN (www.statease.com) [29]. Table 1 lists the ranges of each variable used. Each of the four independent variables had five levels for which the design expert software provided a combination of 30 experiments.

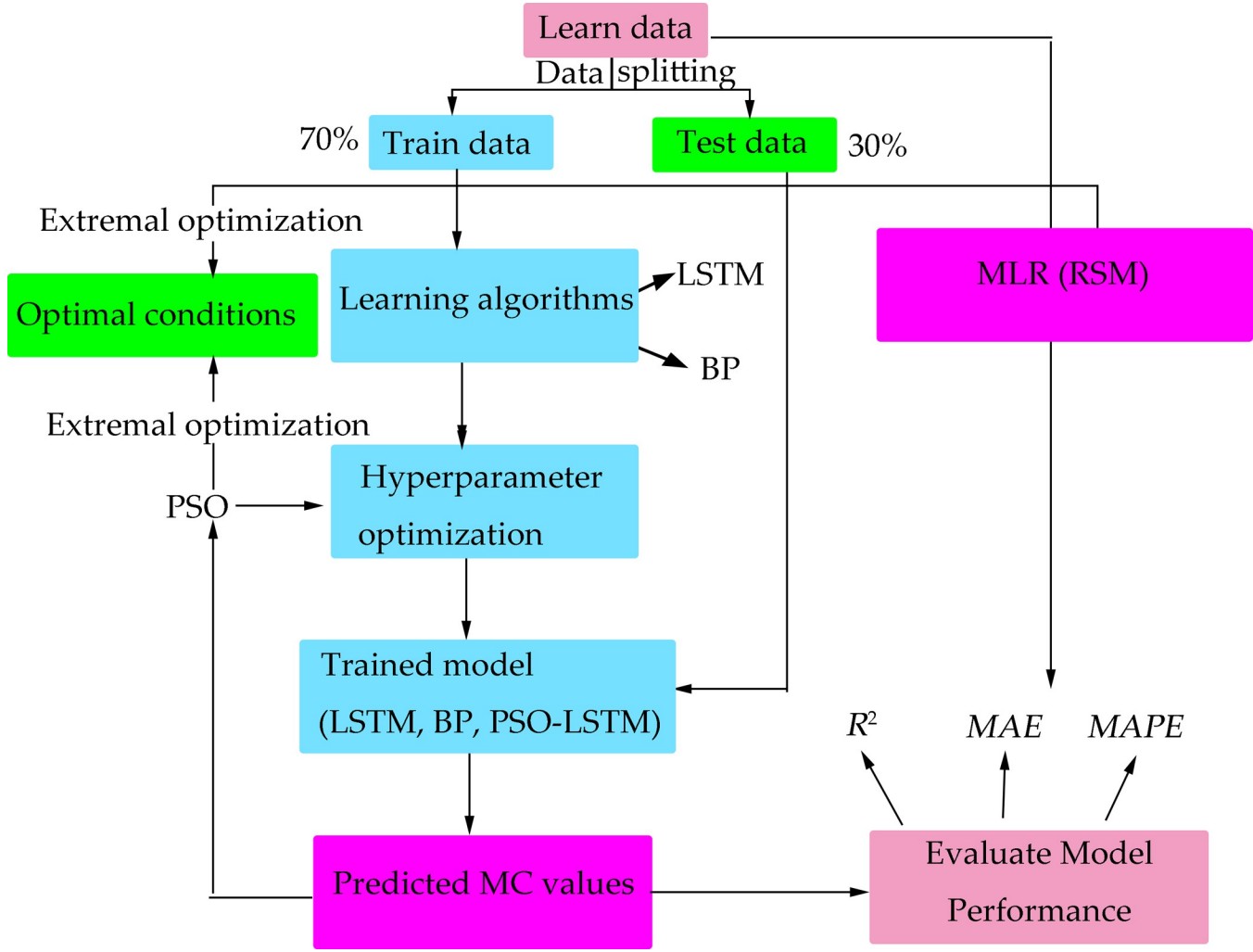

**Fig 5. The main structure of MC prediction model.**

**Table 1. Experimental ranges and levels of independent variables used in multiple linear regressions.**

| Factor | Names | Levels | Range | | | |
|---|---|---|---|---|---|---|
| | | | -1 | 1 | -2 | 2 |
| 1 | Initial MC (%) | 5 | 12.9 | 38.7 | 0 | 51.6 |
| 2 | Inlet air temperature (˚C) | 5 | 180 | 220 | 160 | 240 |
| 3 | Feed rate(kg/h) | 5 | 105 | 135 | 90 | 150 |
| 4 | Inlet air velocity (m/s) | 5 | 10 | 12 | 9 | 13 |

## LSTM modeling

The LSTM is a kind of improved RNN, as it can solve the problem of RNN perception ability decline [30]. In contrast to RNN, the LSTM adds a cell state on its basis, and the LSTM unit controls the cell state through three gates: the forgetting gate, the input gate, and the output gate, as shown in Fig 6 [31].

The internal structure of the LSTM unit is constituted by a sigmoid neural network layer and a point multiplication operation). Whether to discard some information is decided by the sigmoid layer of the forgetting gate. The results of the operation are 1 for reserving and 0 for discarding information. The operation equation of the forgetting gate is

$$f_t = \sigma[W_f \cdot (h_{t-1} \cdot x_t) + b_f], \tag{2}$$

where $f_t$ is the result of the forgetting gate; $W_f$ is the forgetting gate weight matrix; $x_t$ and $h_{t-1}$ are the input of the current time and the output of LSTM at the previous time, respectively; $b_f$ is the bias term of the forgetting gate; and $\sigma$ is the sigmoid activation function.

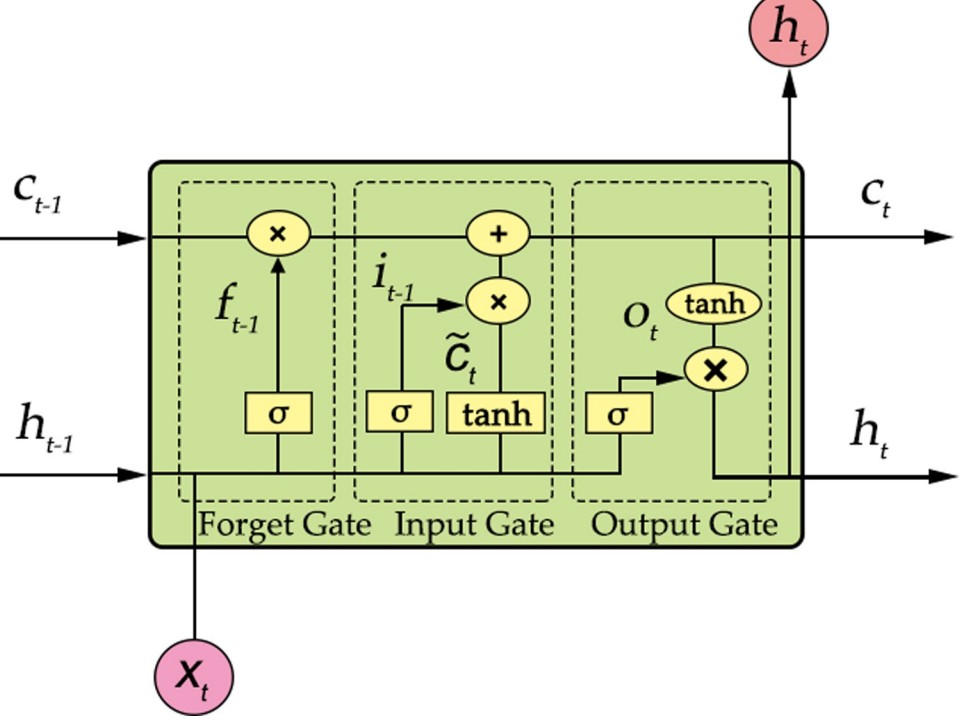

**Fig 6. LSTM unit.**

By adding new information according to the sigmoid layer of the input gate and combining it with the candidate values obtained from the tanh layer, the state update amount is obtained, as shown in Eqs (3) and (4).

$$i_t = \sigma[W_i \cdot (h_{t-1} \cdot x_t) + b_i], \tag{3}$$

$$\tilde{C}_t = tanh[W_c \cdot (h_{t-1} \cdot x_t) + b_c], \tag{4}$$

where $i_t$ is the result of input gate operation; $\tilde{C}_t$ is the candidate value; $W_i$ and $b_i$ are the input gate weight matrix and the bias term, respectively; and $W_c$ and $b_c$ are the weight matrix and the bias term of the element state, respectively.

Considering the information discarded in the forgetting gate, the unit state at the current time can be acquired, as shown in Eq (5).

$$C_t = f_t \cdot C_{t-1} + i_t \cdot \tilde{C}_t, \tag{5}$$

where $C_t$ is the cell state at the current time; $C_{t-1}$ is the unit state of the previous time; $f_t \cdot C_{t-1}$ shows the discarded information; and $i_t \cdot \tilde{C}_t$ refers to the state update quantity.

The sigmoid layer of the output gate determines which information to output and then combines the candidate cell state processed by the tanh layer to obtain the output, as shown in Eqs (6) and (7).

$$O_t = \sigma(W_o \cdot [h_{t-1} \cdot x_t] + b_o), \tag{6}$$

$$h_t = O_t \cdot \tanh(C_t), \tag{7}$$

where $O_t$ is the operation result of the output gate; $h_t$ is the output of the current time; and $W_o$ and $b_o$ are the weight matrix and the bias term of the output gate, respectively.

## Optimizing LSTM prediction model by PSO

To make the prediction model better match the data characteristics of MC under different working conditions, PSO algorithm was used to optimize the LSTM model, and PSO-LSTM model was constructed to obtain better parameter combination. Firstly, the batch size and the number of hidden layer units were randomly initialized within a given range as the initial parameters of the LSTM model. The initial model and the trained model were trained and predicted, respectively, through the divided training data and verification data, and the average absolute percentage error of the prediction results was taken as the fitness function $f$. The fitness function $f$ is defined as

$$f = \frac{1}{N}\sum_{m=1}^{N}\frac{\hat{y}_m - y_m}{y_m}, \tag{8}$$

where $\hat{y}_m$ is the m-th tag value, and $y_m$ is the m-th predicted value.

The number of iterations was 500, and the inertia constant was 0.7. When the number of particle iterations reached 500 or the fitness value reached the set requirements, the iteration was stopped.

## Results

### Data preprocessing

In this stage, the neural network model was studied and constructed under the TensorFlow learning framework based on Python. The TensorFlow 2.0 library was loaded into Anaconda

in advance, and then the NumPy, Pandas, and Matplotlib libraries in Python data analysis were imported. In the data preprocessing stage, the original data obtained by the RSM experiment design were simply processed, and the data set was divided into the training set and the test set. The training data for model training accounted for 70.4%, and the test data covered 29.6% to test the generalization error of the model.

To increase the speed and accuracy of the neural network gradient descent to find the optimal solution, the input data were normalized, and all the numerical information was gathered within the range 0–1 [32]. After the model construction, denormalization was carried out; the normalization equation is

$$x = \frac{I_i - I_{min}}{I_{max} - I_{min}}, \tag{9}$$

where $x$ is the normalized input value, $I_i$ is the sample data value before normalization, $I_{min}$ is the minimum value in the sample data, and $I_{max}$ is the maximum value in the sample data.

## Model evaluation index

In this study, the mean absolute error (MAE), mean square error (MSE), mean absolute percentage error (MAPE), Pearson correlation coefficient, $r$, and determination coefficient, $R^2$, between predicted data and real data were chosen as the evaluation indexes of model performance.

MAE accurately reflects the error between the predicted value and the real value. The smaller the MAE is, the closer the predicted value is to the real value, and the more accurate the prediction is. MAE is expressed by Eq (10).

$$MAE = \frac{1}{n} \sum_{i=1}^{n} |y_i - \tilde{y}_i| \tag{10}$$

MSE shows the difference between the predicted value and the real value. The smaller the MSE value is, the smaller the difference between the predicted value and the real value is, and the better the accuracy of the model is. MSE is expressed by Eq (11).

$$MSE = \frac{1}{n} \sum_{i=1}^{n} (y_i - \tilde{y}_i)^2 \tag{11}$$

MAPE indicates the percentage of relative error between the predicted value and the real value. The smaller the MAPE value is, the better the model is. MAPE is expressed by Eq (12).

$$MAPE = \frac{100\%}{n} \sum_{i=1}^{n} \left| \frac{y_i - \tilde{y}_i}{y_i} \right| \tag{12}$$

Pearson correlation coefficient, $r$, refers to the linear correlation between the predicted value and the real value. The closer $r$ is to 1, the better the correlation between the predicted value and the real value is. $r$ is expressed by Eq (13).

$$r = \frac{COV(Y, \tilde{Y})}{\sqrt{VAR(Y) \cdot VAR(\tilde{Y})}} \tag{13}$$

The determination coefficient, $R^2$, displays the reliability of the model. The closer $R^2$ is to 1, the more reliable the model is. $R^2$ is expressed as Eq (14).

$$R^2 = 1 - \frac{\sum_{i=1}^{n} (y_i - \tilde{y}_i)^2}{\sum_{i=1}^{n} (\tilde{y}_i - y_{i.ave})^2}, \tag{14}$$

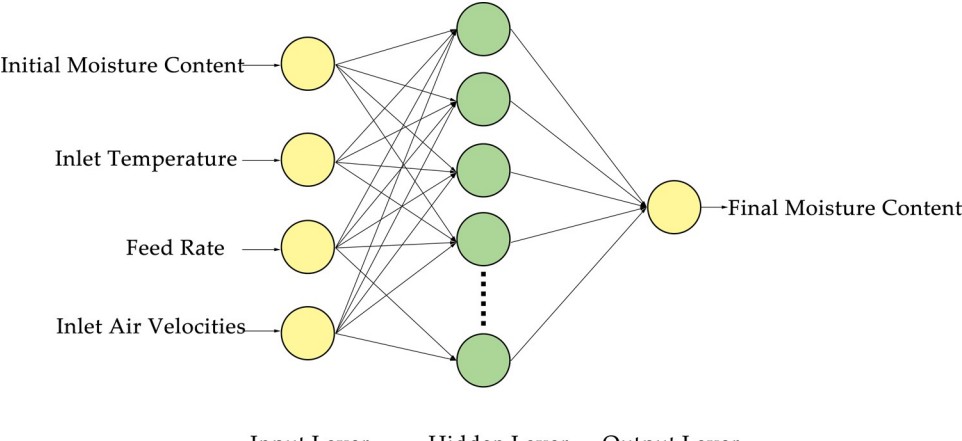

**Fig 7. BP structure of wood fiber moisture content prediction.**

where $n$ is the number of test data sets, $y_i$ is the true value of the i-th sample point, $\tilde{y}_i$ is the predicted value of the i-th sample point, $y_{i.ave}$ is the average value of the sample real value, $Y$ is the true value of the sample, and $\tilde{Y}$ is the predicted value of the model. Further, $\text{COV}(Y, \tilde{Y})$ indicates the covariance of $Y$ and $\tilde{Y}$, $\text{VAR}(Y)$ denotes the variance of $Y$, and $\text{VAR}(\tilde{Y})$ represents the variance of $\tilde{Y}$.

## Model parameter setting

In this study, Python programming language with Tensorflow 2.0 framework was used to build the BP neural network (Fig 7). It can be noticed that a three-layer structure was adopted in the network, and there were four input layer nodes: the initial MC, inlet air temperature, feed rate, and inlet air velocity. Besides, there were 50 hidden layer nodes and one output node, which corresponds to the final MC obtained from the experiment. The sigmoid function was selected to activate the hidden layer, while the learning rate was set to 0.001 and the number of iterations to 1000. Furthermore, the adaptive motion estimation optimizer was adopted to update the weights of the neural network [33].

As shown in Fig 8, the framework of the multivariable LSTM prediction model can be divided into three parts: the input layer, the LSTM layer, and the output layer. The input of the input layer belonging to the LSTM model is $I = (I_1, I_2, I_3, I_4)$. After normalization and weighting, the input unit becomes the input $x_i$ of the LSTM layer. The number of neural units in the LSTM layer is 50. After processing, the output of the LSTM layer is $H_i$, and the output result of the LSTM layer becomes the input of the output layer after weighted processing. Following the fully connected layer, the output data are denormalized to get the predicted value of the final MC, $y$. When the error between the actual output and the expected output exceeds the specified accuracy, it enters the error BP stage. The output layer corrects the weight of each layer by decreasing the error gradient, and the error propagates back to the LSTM layer and the input layer.

## Single-factor results

The effects of different process parameters on the final MC of WF were studied. The feeding rate was fixed at 120 kg/h, the inlet air velocity was 11 m/s, and the inlet temperature was changed to 120˚C, 140˚C, 160˚C, 180˚C, 200˚C, 220˚C, and 240˚C. In the same way, the

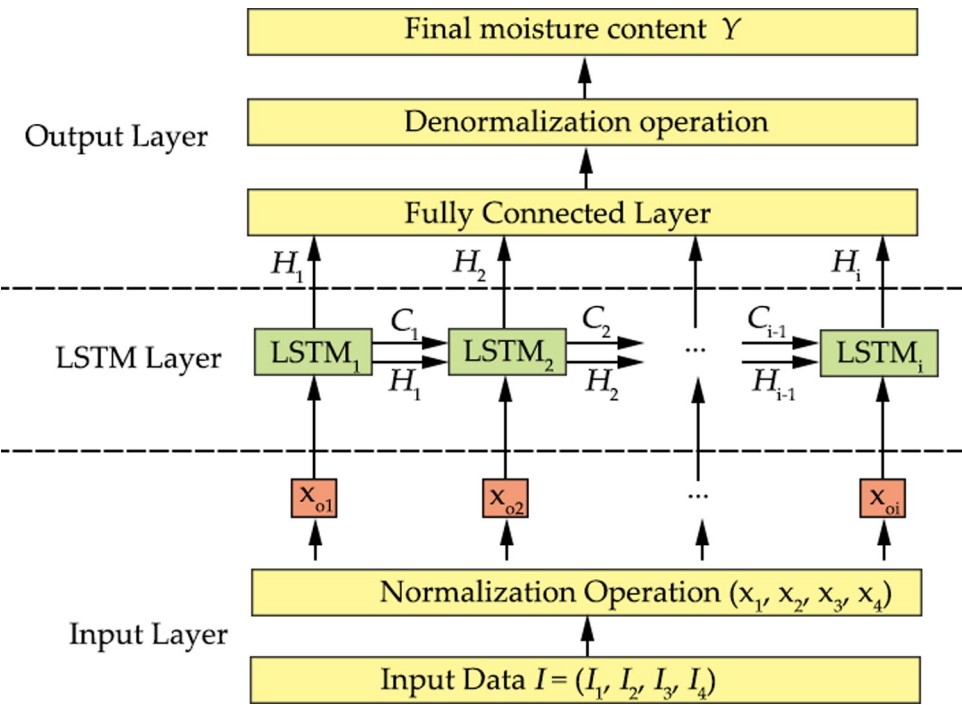

**Fig 8. LSTM structure of wood fiber moisture content prediction.**

feeding rate was fixed at 120 kg/h, the inlet temperature was 200˚C, and the inlet air velocity was changed to 7 m/s, 8 m/s, 9 m/s, 10 m/s, 11 m/s, 12 m/s, and 13 m/s. Similarly, the inlet air velocity was fixed at 11 m/s, the inlet temperature was 200˚C, and the feed rate was changed to 60 kg/h, 75 kg/h, 90 kg/h, 105 kg/h, 120 kg/h, 135 kg/h, and 150 kg/h. The experiment was repeated three times, and each index was determined three times under the same conditions.

Fig 9(A) shows how the final MC of poplar fiber changes with the inlet air temperature. Obviously, the higher the inlet air temperature of the dryer is, the faster the water molecules move, which is conducive to the vaporization of the fiber surface and increases the temperature and humidity gradient inside and outside the fibers. During the experiment, the drying temperature should be increased to obtain the lowest final MC. Fig 9(B) illustrates how the final MC of poplar fiber changes with the inlet air velocity. With an increase in the inlet air velocity, the final MC shows an upward trend and then a downward trend. Fig 9(C) demonstrates how the final MC of poplar fiber changes with the feed rate. The final MC shows an upward trend with an increase in the inlet air velocity, mainly because the total evaporation of fiber water increases and the air temperature decreases with an increase in feed rate. In this case, the heat and mass transfer force and the mass transfer rate decrease.

## Operation and verification of ICD system

In this drying system, the anemometers were set in the impulse dryer to measure the straight pipe air velocity and impulse pipe air velocity, and the monitoring system recorded the results when it was stable. During the experiment, the temperature of the dryer was 160˚C, and the frequency of the induced draft fan was 0–50 Hz. When processing the data, the blank air velocity was subtracted when the induced draft fan did not start from the data result, and the arithmetic mean value of five measurements was taken to obtain the air velocity value. When adding materials, the results were recorded in the steady state.

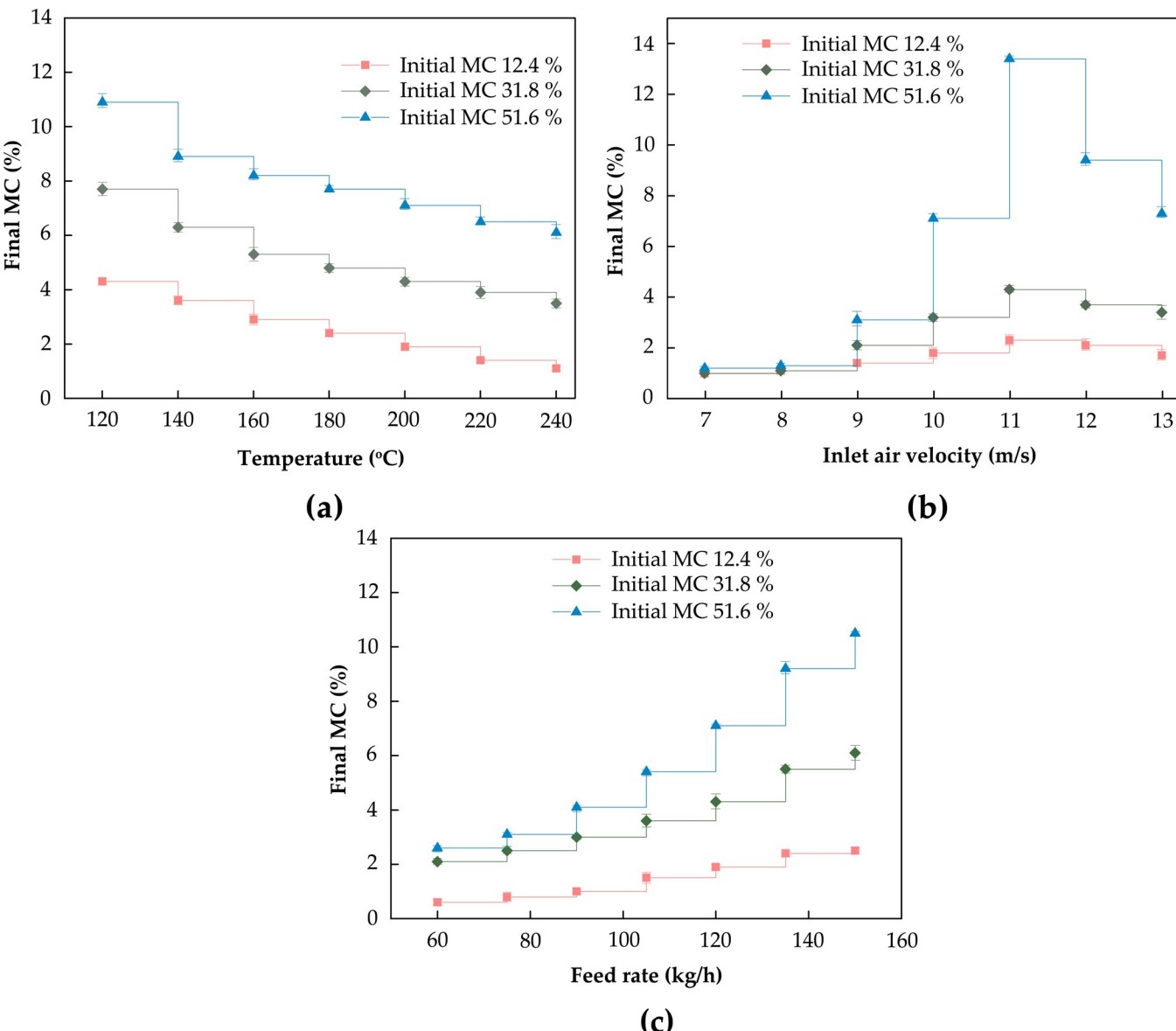

**Fig 9.** Results of the single factor test, (a) effects of different inlet air temperatures on final moisture content, (b) effects of different inlet air velocities on final moisture content, (c) effects of different feed rates on final moisture content.

In Fig 10, it can be noticed that, under the same induced draft fan frequency, the air velocities of straight pipe and impulse pipe are positively correlated with the frequency of induced draft fan. In particular, the greater the frequency of induced draft fan is, the greater the air velocities of straight pipe and impulse pipe are. According to the experimental results, under the same induced draft fan frequency, the air velocity of the impulse pipe was lower than that of the straight pipe, which indicated that the airflow slowed down in the impulse pipe. The higher the MC of the added material is, the more obvious the decrease in the air velocities of the straight pipe and impulse pipe is; hence, the greater the MC is, the greater the moisture in the air is, and the more obvious the decrease in the air velocity is.

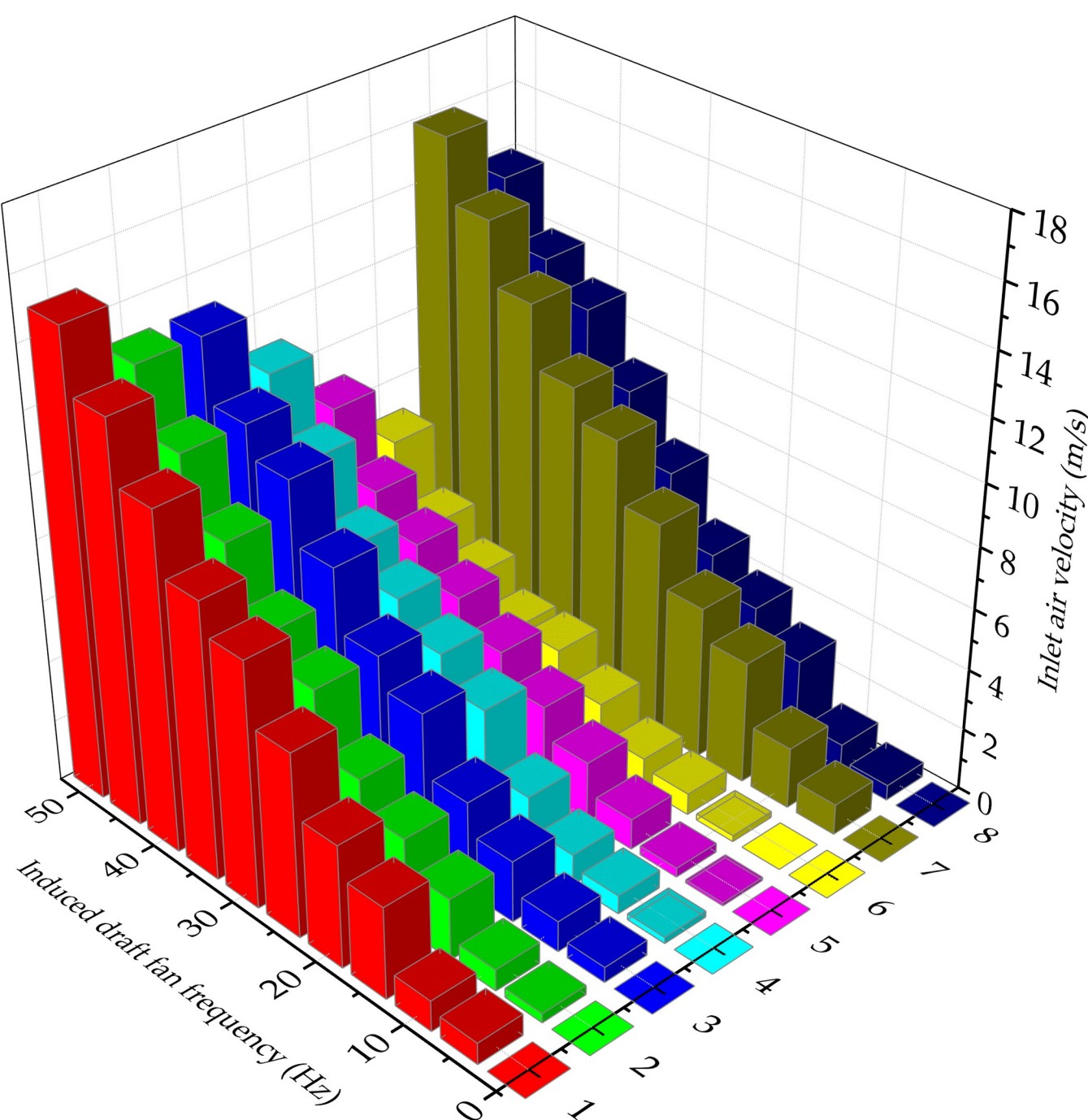

**Fig 10. Air velocity value corresponding to different frequency of the dryer.** (1) Straight pipe airflow velocity of 10 wt% wood fibers, (2) Impulse pipe airflow velocity of 10 wt% wood fibers, (3) Straight pipe airflow velocity of 30 wt% wood fibers, (4) Impulse pipe airflow velocity of 30 wt% wood fibers, (5) Straight pipe airflow velocity of 50 wt% wood fibers, (6) Impulse pipe airflow velocity of 50 wt% wood fibers, (7) No-load straight pipe airflow velocity, (8) No-load impulse pipe airflow velocity.

## Different drying methods on MC uniformity

By comparing the results of ICD, oven-drying, and rotary drum-drying [34], 30 samples were randomly taken from the dried WF to investigate the final MC. The data were sorted by Excel

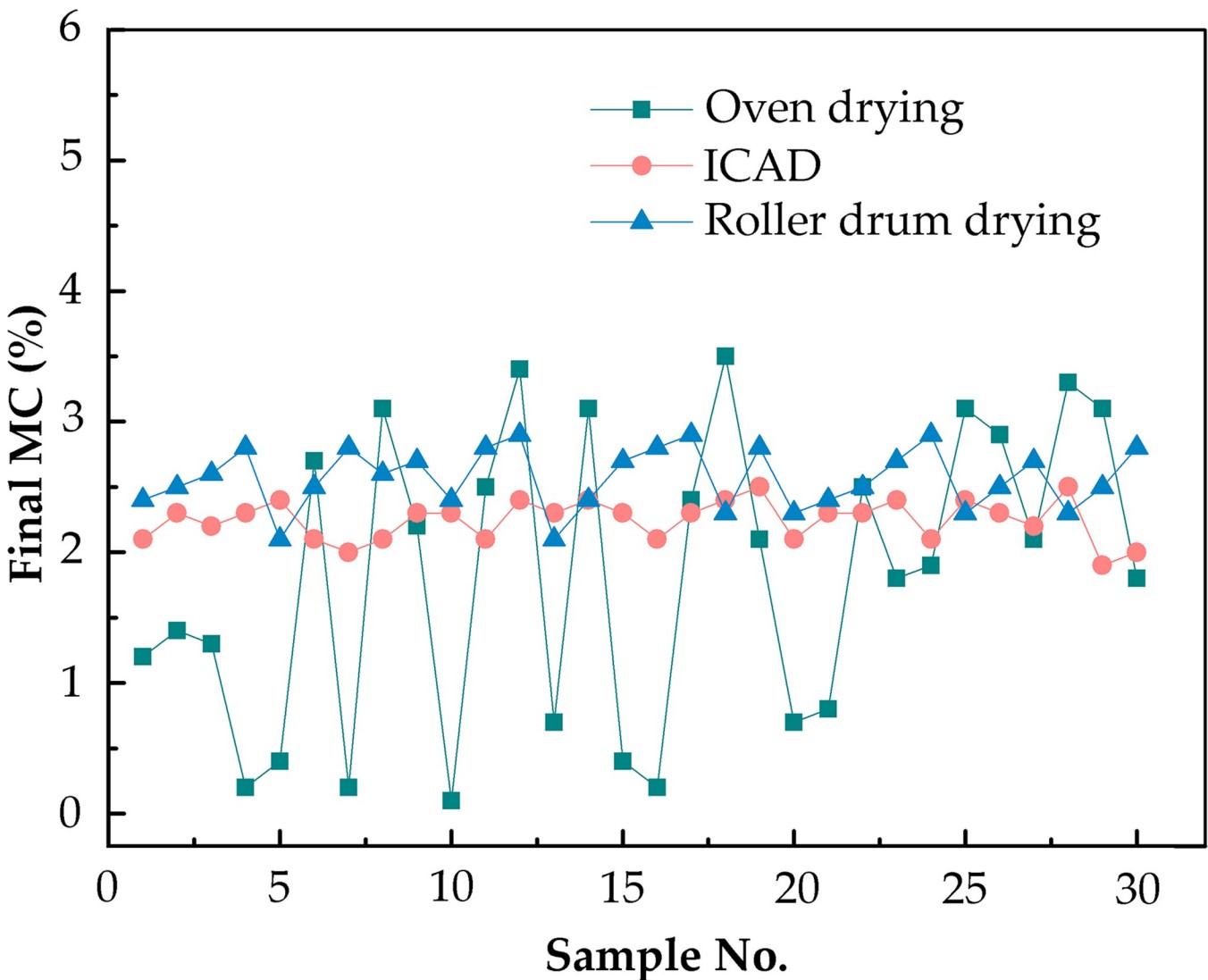

**Fig 11. Final moisture content by different drying method.**

and subjected to analysis of variance (ANOVA) using SPSS 7.05 data processing system (IBM Inc., USA). The results are presented in Fig 11 and Table 2. According to the results, the MC of WF dried by ICD was more uniform than that dried by the other two drying methods. This indicated that, during the stable discharging stage, WFs were saturated in gas phase and had higher MC in air, which affected mass transfer efficiency to a certain extent. The final MC of WF could reach 1–3% final MC of WPCs.

**Table 2. The final moisture content of Poplar fiber by different drying methods.**

| Drying method | Maximum (%) | Minimum (%) | Average value (%) | Standard deviation | Variation coefficient | Drying time (min) |
|---|---|---|---|---|---|---|
| *Oven drying* | 3.12 | 0.23 | 1.41 | 1.01 | 0.7163 | 360 |
| *Roller drum drying* | 2.92 | 2.12 | 2.44 | 0.27 | 0.1043 | 60 |
| *ICAD* | 2.44 | 2.02 | 2.13 | 0.17 | 0.0798 | 5.2 |

**Table 3. Optimum operating parameters of different drying methods.**

|  | ICD | Oven drying | Rotary drum drying |
|---|---|---|---|
| Manufactor | Northeast Forestry University | Changzhou Lehua Drying Co., Ltd., Jiangsu, China | Shandong Hanyu Environmental Protection Equipment Co., Ltd., Shandong, China |
| Model | MQG-50 | CT/CT-I | HYHG1.2x12 |
| Wet fiber capacity | 120kg | 100kg | 120kg |
| Operating equipment power | 46.2Kw (Electric heater); 8.8Kw (Induced draft fan motor) 0.88Kw (Screw charger motor) 0.75Kw (YCD-HX unloader) | 15Kw | (1)18.75Kw (Motor power) (2) 75Kw (Heater power) |
| Processing time | 1h | 6h | 1h |

## Energy consumption cost comparison of different drying methods

The fiber with an initial MC of about 20% was mainly used to observe the energy consumption of different drying methods to select a more energy-saving and efficient fiber drying process. The temperature of ICD was 220˚C, inlet airflow was 11 m/s, feed rate was 120 kg/h. The energy consumption was under the best operating conditions determined in the experiment, and the electric heater was heated by two groups of heating tubes. Owing to the existence of automatic switch, according to the calculation, the electric loss was 70% of the actual electric quantity at 220˚C.

Table 3 shows the optimum operating parameters according to the pilot plant established in the experiment. Taking drying 1000 kg wet fiber as an example, the energy consumption costs of the three methods were calculated, as shown in Table 4 (in which the electricity charge was 0.7 CNY/kWh according to the industrial electricity in China). It can be noticed that the energy consumption cost of ICD per 1000 kg of wet fiber was 329 CNY, which was less than that of oven-drying and rotary drum-drying, proving the superiority of ICD. The equipment was obtained on the basis of a pilot test. After engineering application in the future, the process intensification could be further improved, and the energy consumption cost could be further reduced.

## MLRs

The results of the RSM experiment were analyzed by Design Expert 10.0.4, and the data of the RSM experiment were fitted with the multiple regression model analyzed by ANOVA. The results of ANOVA are shown in Table 5. The $F$-value of the model is 71.86, and the $P$-value is lower than 0.0001, indicating that the model is significant.

The effects of the initial MC (A) and inlet air temperature (B) on the final MC were extremely significant ($P < 0.0001$), that of feed rate (C) was generally significant, and that of inlet velocity (D) on deposition rate was not significant. By comparing the mean square values, it can be inferred that the order of process parameters affecting the final MC of WF is A > B > C > D. $A^2$ is a significant factor in the quadratic term, and the rest are not significant. The $P$-value of the lack of fit factors is 0.2762 ($P > 0.05$), revealing that the lacking fit factor was not

**Table 4. Energy consumption cost of different drying methods.**

|  | ICD | Oven drying | Rotary drum drying |
|---|---|---|---|
| Manufactor | MQG-50 | CT/CT-I | DLSG1409 |
| Time for drying 1000kg | 8.3h | 60h | 8.3h |
| Hourly power | 56.63Kw | 15Kw | 75Kw |
| Total power | 470Kw·h | 900Kw·h | 622.5Kw·h |
| Total energy consumption cost | 329 CNY | 630 CNY | 437.5 CNY |

**Table 5. Variance analysis of regression model.**

| Source | Sum of Squares | Degree of freedom | Mean Square | F value | P-value | |
|---|---|---|---|---|---|---|
| Model | 88.15 | 14 | 6.30 | 71.86 | <0.0001 | Significant |
| A | 110.08 | 1 | 110.08 | 164.06 | <0.0001 | |
| B | 21.62 | 1 | 21.62 | 32.22 | <0.0001 | |
| C | 6.00 | 1 | 6.00 | 8.94 | 0.0075 | |
| D | 0.81 | 1 | 0.81 | 1.20 | 0.2866 | |
| AB | 4.41 | 1 | 4.41 | 6.57 | 0.0190 | |
| AC | 0.72 | 1 | 0.72 | 1.08 | 0.3125 | |
| AD | 0.040 | 1 | 0.040 | 0.060 | 0.8097 | |
| BC | 0.063 | 1 | 0.063 | 0.093 | 0.7635 | |
| CD | 2.500E-003 | 1 | 2.500E-003 | 3.726E-003 | 0.9520 | |
| A2 | 0.90 | 1 | 0.90 | 20.63 | 0.0003 | |
| B2 | 0.072 | 1 | 0.072 | 0.78 | 0.3784 | |
| C2 | 0.025 | 1 | 0.025 | 0.72 | 0.4253 | |
| D2 | 1.27 | 1 | 1.27 | 13.38 | 0.0014 | |
| Residual | 12.75 | 19 | 0.67 | | | |
| Lack of Fit | 10.60 | 14 | 0.76 | 1.76 | 0.2762 | Not Significant |
| Pure Error | 2.15 | 5 | 0.43 | | | |
| Cor Total | 162.26 | 29 | | | | |

significant, which proves that the model could better describe the relationship between the final MC and various factors. Among them, the determination coefficient, $R^2$, of the RSM model is 0.8801. The final quadratic polynomial model of response regression prediction is established by Eq (15).

$$Y = 4.31 - 0.95A + 0.18B + 0.50C + 2.14D - 0.60AB - 0.062AC - 0.52AD - 0.012BC \\ - 0.050BD + 0.21CD - 0.005A^2 + 0.001B^2 - 0.009C^2 - 0.004D^2 \qquad (15)$$

To determine the best fitting degree of the selected model, the normal distribution map of the residual error is taken from the Design Expert software (Fig 12). According to the normal probability distribution map, the maximum final MC data point falls in the straight line of the normal distribution of the response data. This means that the response data set, i.e., the final MC, is normally distributed relative to the proposed linear model.

By using the model graph option in the analysis module, the contour lines and response surface diagrams for evaluating the interaction strength of various experimental factors can be obtained. Thus, they can be used to predict the interaction of variables, so that the optimal drying process parameters can be determined. The interaction between various factors is shown in Fig 13.

As shown in Fig 13(A), with a decrease in the initial MC (A) and an increase in the inlet air temperature (B), the final MC decreased significantly, i.e., the initial MC and inlet air temperature jointly affect the final MC. The bulge on the response surface is more intuitive in Fig 13 (A) than in Fig 13(B), which shows that the influence of inlet air temperature on the final MC is more significant. The reason is that the higher the inlet air temperature is, the larger the temperature gradient difference will be. Therefore, the speed of moisture migration inside the fiber to the surface will increase, further resulting in the lower final MC of the fiber. Fig 13(B) illustrates that, with a decrease in the initial MC and feed rate, the final MC decreases dramatically, since fewer materials are put into the pipeline simultaneously, and the MC of the airflow in the pipeline is less, which improves the MC gradient inside the fiber and on its surface. The

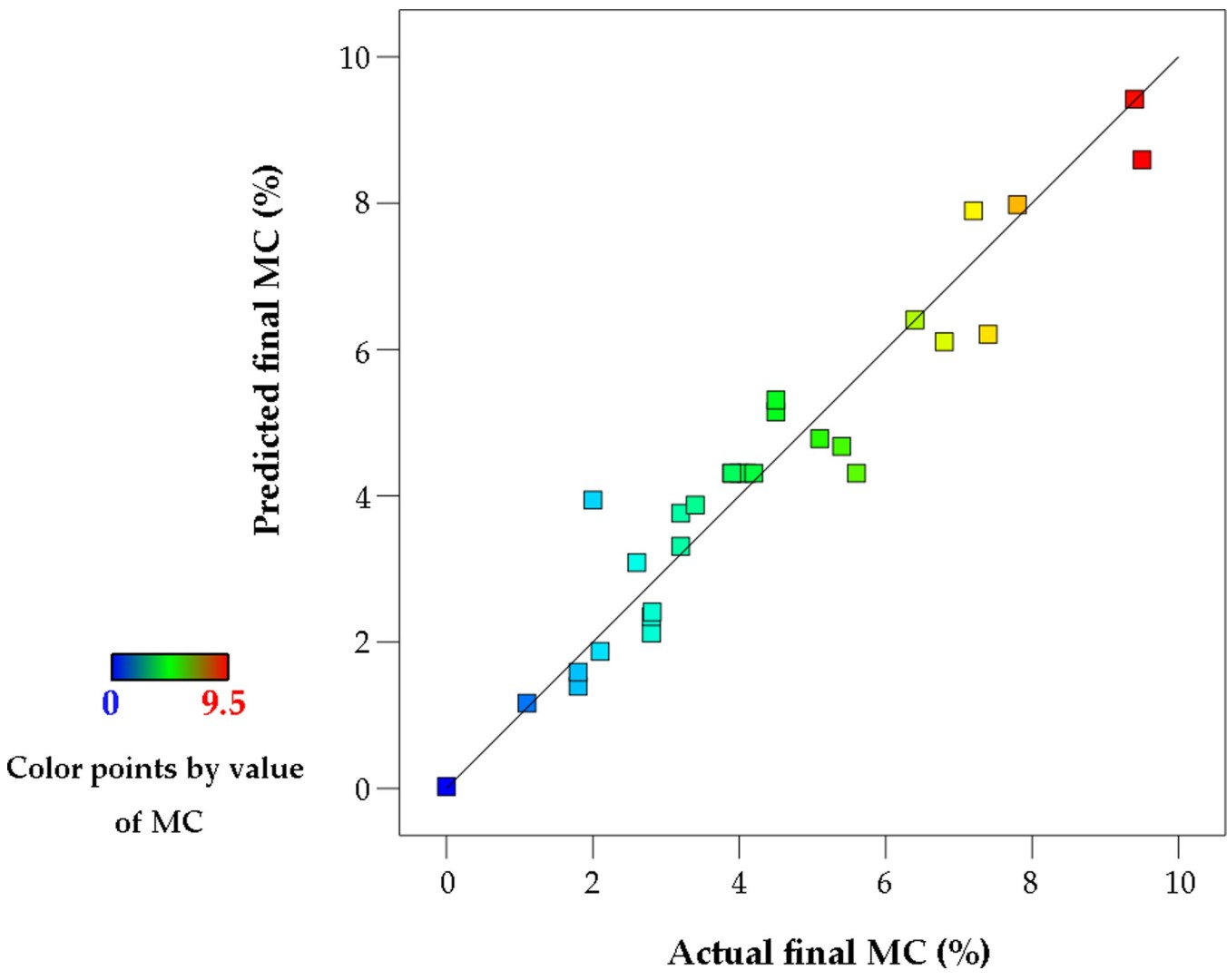

**Fig 12. Predicted versus actual final MC values.**

moisture migration speed increases, and the final MC decreases. Therefore, among all the influencing factors, changing the initial MC has the most obvious influence on the final MC, and the convex radian presented on the response surface diagram is most intuitive.

## LSTM results

Taking the Design Expert's experimental design parameters and experimental results as training samples, the program was developed in the Spyder environment of Anaconda software. The initial weights and thresholds were assigned to the BP neural network and the LSTM for learning and updating, and the final MC prediction model was established. The iteration number of the initial network parameters was set to 1000, while the learning rate was set to 0.001. Besides, 30 groups of data in the sample were selected as training data, and 12 groups were used as test data to evaluate the neural network model. By adopting different neural network models to train the model on the training set and test on the test set, the performance of different neural network models under different parameter settings was examined.

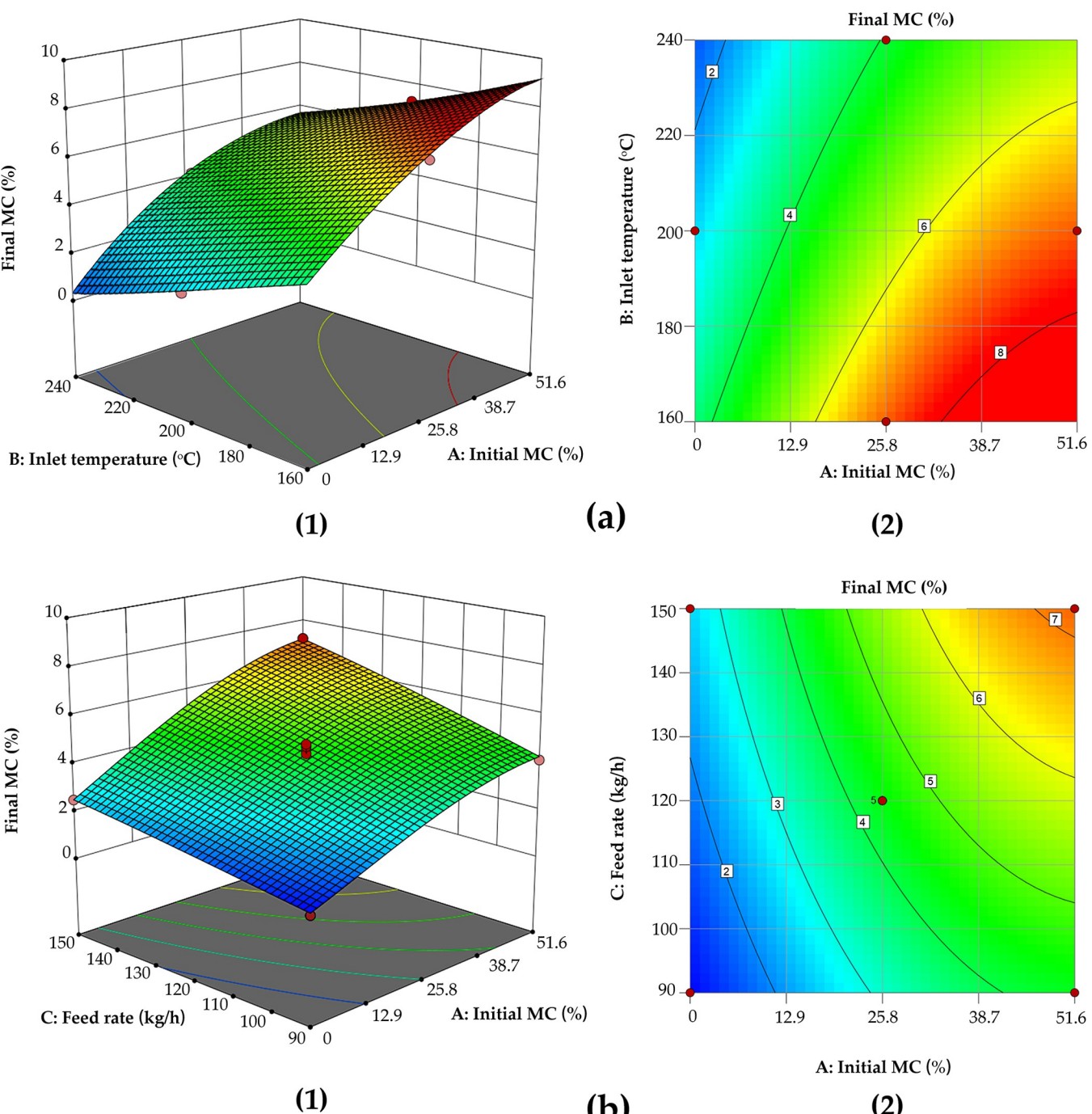

**Fig 13.** 3D Response surface (1) and contour map (2) concerning the interaction of experimental factors, (a) interaction between inlet air temperature and initial moisture content, (b) interaction between feed rate and initial moisture content.

The regression analysis of the final MC was carried out on the test sets of RSM, the BP neural network model, and the LSTM. Fig 14 shows the comparative analysis on the experimental predicted values of the three models. The evaluation indexes of RSM, the BP model, and the LSTM model are obtained by calculation. The results are shown in Table 6.

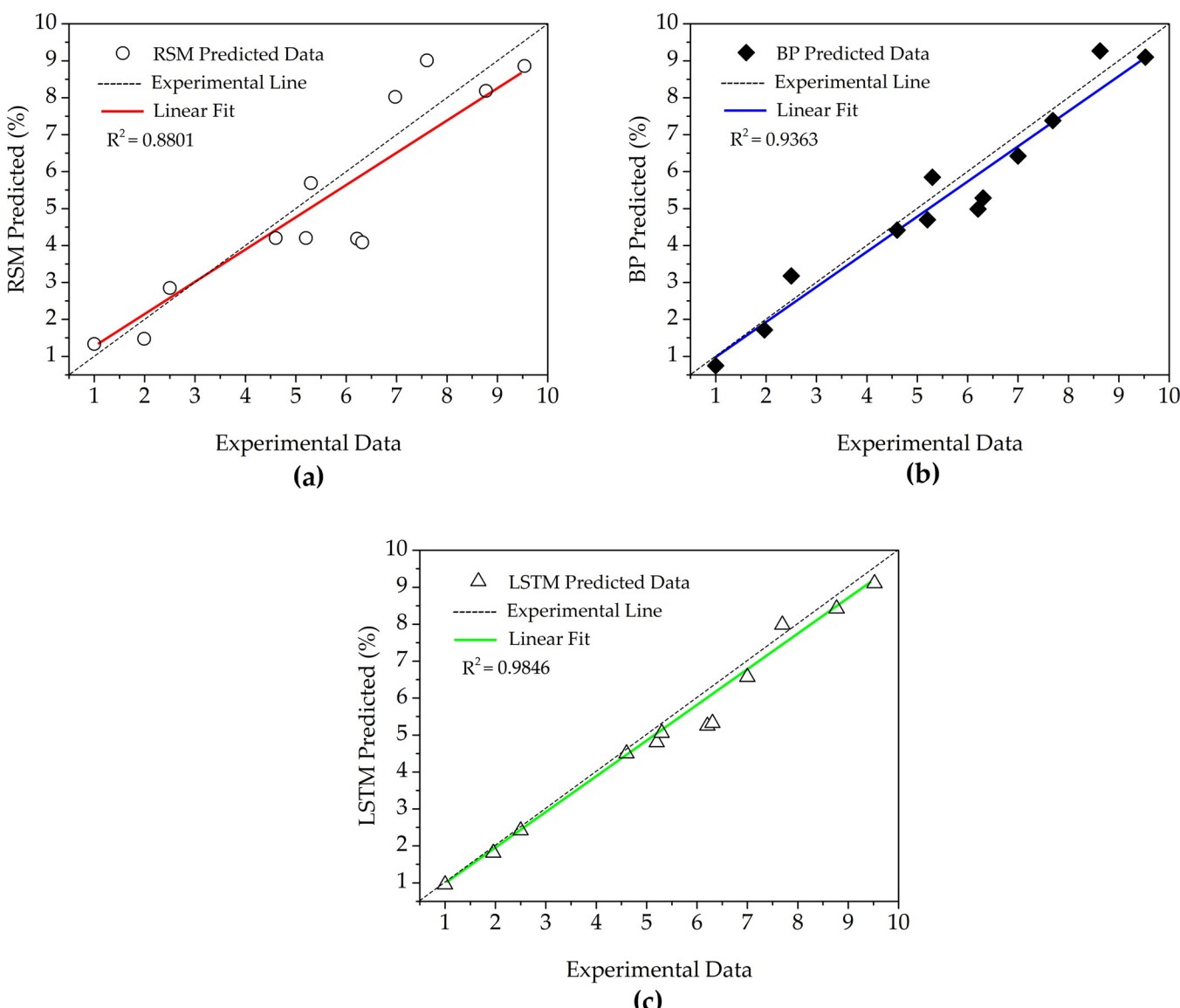

**Fig 14.** Comparative analysis on experimental predicted values of three models, (a) RSM model, (b) BP model, (c) LSTM Model.

## Particle Swam Optimization (PSO) to optimize processing parameters

PSO is a computational model to search for the optimal solution by simulating the natural evolution process. The extremum optimization of PSO takes the predicted result of the trained neural network as the individual fitness value and searches for the global optimal value and corresponding input value of the function through selection, crossover, and mutation

**Table 6. Prediction effect of RSM, BP neural network and LSTM neural network on the moisture content of wood fiber ICD.**

| Model | $R^2$ | $r$ | MSE | MAE | MAPE |
|---|---|---|---|---|---|
| RSM | 0.8801 | 0.8715 | 0.3647 | 0.3911 | 0.6876 |
| BP | 0.9363 | 0.9350 | 0.1665 | 0.3020 | 0.4370 |
| LSTM | 0.9446 | 0.9809 | 0.2316 | 0.2207 | 0.4124 |

**Table 7. Comparison and verification of different methods for Optimizing ICD.**

| Model | Initial MC (%) | Inlet air temperature (˚C) | Feed rate (kg/h) | Inlet air velocity (m/s) | Predicted final MC (%) | Actual final MC (%) | Relative error (%) |
|---|---|---|---|---|---|---|---|
| *RSM* | 12.4 | 200 | 90 | 9 | 1.00 | 1.33 | 33.0 |
| *BP- PSO* | 10.6 | 238 | 90 | 9 | 1.12 | 1.43 | 27.7 |
| *LSTM-PSO* | 10.3 | 242 | 90 | 8 | 1.03 | 0.96 | 6.8 |

operations. The parameters set in the PSO were a number of iteration evolution of 100 times, a population size of 20, a crossover probability of 0.4, a mutation probability of 0.2, and an individual length of 1. The results are shown in Table 7. The optimal process parameters of the BP-PSO model were an initial MC of 10.6%, an inlet air temperature of 238˚C, a feed rate of 90 kg/h, and an inlet air velocity of 9 m/s. The optimal process parameters of the LSTM-PSO model were an initial MC of 10.3%, an inlet air temperature of 242˚C, a feed rate of 90 kg/h, and an inlet air velocity of 8 m/s. The optimal process parameters obtained by the three methods were compared.

In the optimization problem, response surface analysis can only focus on factors and levels on the drying process at the known factor level, thus the optimization results are not global. BP and LSTM, as intelligent algorithms, can perform global optimization with the combination of PSO, which is global and scientific, and the accuracy of the optimization results is positively related to the accuracy of the selected neural network.

## Prediction of PSO-LSTM combined model with expanded sample size

The PSO algorithm first initializes a group of particles in the feasible solution space and employs three indicators—position, speed, and fitness—to represent the characteristics of each particle; the fitness value is used as the standard to measure the quality of particles. In this paper, the particle position corresponds to the initial MC, inlet temperature, feed rate, and inlet velocity, and the fitness value corresponds to the final MC. Particles move in the solution space and update individual positions by tracking individual and global optimal positions. Besides, the value is calculated every time the particle updates the position, and the individual and global optimal positions are updated by comparing the fitness value of the new particle with that of the individual and global optimal positions. The process of PSO is shown in Fig 15.

A total of 400 groups of sample data were expanded, 304 of which were used for model training and the remaining 96 for prediction. PSO algorithm was used to optimize the number of hidden layer neurons and the learning rate of LSTM model; it was also used to judge whether to update the individual and global optimal solutions until the termination conditions were met to obtain the optimal parameters. In the training process, the optimized LSTM

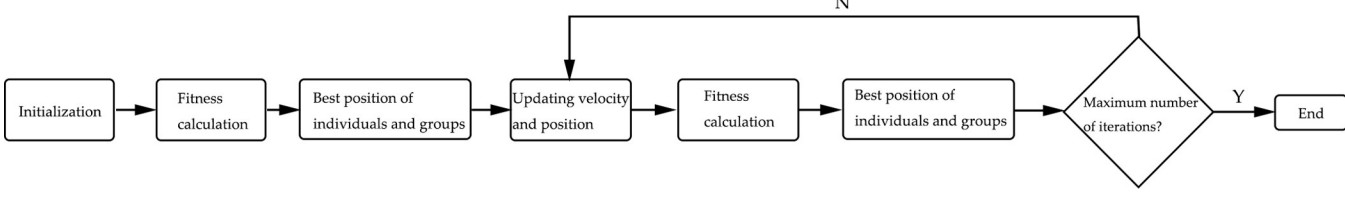

**Fig 15. PSO algorithm process.**

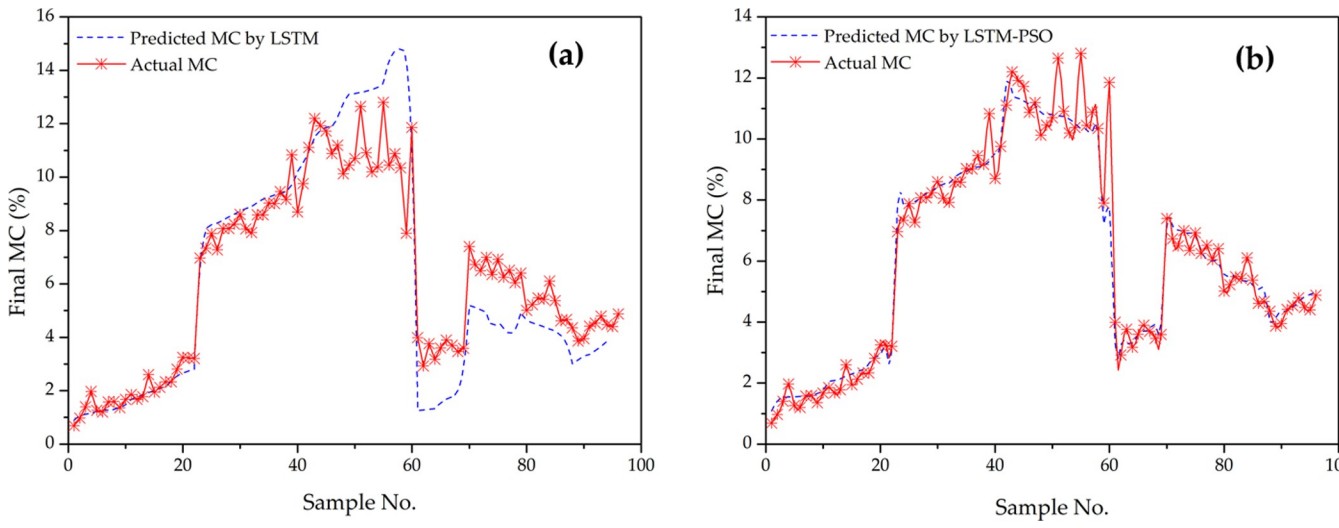

**Fig 16. Actual value and predicted value of the LSTM and LSTM-PSO.**

model was obtained by calculating the particle fitness function and updating the particle position simultaneously. The LSTM and PSO-LSTM prediction models are shown in Fig 16.

## Discussion

### Effect of different drying methods on MC uniformity

Among the different drying methods, the ICD had the lowest variable coefficient of 0.0798 proving the best MC uniformity (Table 2). The lightweight fibers in the ICD system were discharged from the system first, whereas the heavier fibers were repeatedly distributed and arranged in the system, and the drying time was longer, thus the MC of the fibers was more uniform. In addition, although the variable coefficient of the rotary drum-drying method (0.1043) was slightly higher than that of the ICD method (0.0798), the rotary drum-drying method took a longer time. Oven-drying, although a common method, resulted in the poor uniformity of the fiber MC (Fig 11). The lower MC could easily cause fracture of fibers when fiber and plastic were mixed, resulting in lower mechanical properties of WPCs. One possible reason is that the heat and mass transfer efficiency of different parts of the exposed accumulated fibers was different. In addition, the exposure of poplar fibers in hot air was less, and the bound water was difficult to discharge from the fibers. Therefore, Uniform MC could be obtained more easily by ICD method and drum-drying method, and the drying time of ICD was shorter than that of drum-drying.

### Relationship between drying parameters and final MC

The temperature of airflow is the main external factor in determining the drying efficiency [35]. With an increase in the inlet air temperature, the drying rate of WFs increased. The higher the temperature was, the shorter the time to reach the required final MC was. Thus, the evaporation speed of moisture on the surface of WF increased with air temperature. Because of the increase in MC gradient and temperature gradient inside and outside WF, the diffusion speed of moisture in WF increased. Therefore, increasing the airflow temperature was conducive to increasing the drying speed of WFs. The higher the air temperature was, the higher the

precipitation rate was, but the higher the energy consumption was. The temperature of WFs and the temperature of moisture in fibers increased with the increase in airflow temperature.

WF drying is the process of using heat energy to remove water from fibers, thus the MC of WFs decreases continuously. According to the variance analysis (Table 5), the influence of the initial MC on the final MC of fibers was extremely significant. The initial MC played an important role in fiber drying process. The initial MC of wet fibers should be uniform in the drying process, otherwise, the process parameters were very difficult to control, which would lead to the unqualified final MC. Secondly, the low initial MC was better for drying, so that the drying time was short and the energy consumption was low. WFs used in WPCs come from a wide range, most of which are low-value wood, processing residues, and waste wood with a wide range of initial MCs. The final MC of WPCs should be in the range 1–3% to ensure sufficient fluidity in the composite process without many pores, so that the produced composites can meet the national standards of China [36]. When the MC was lower than the fiber saturation point, there was a noncrystalline region between water and cellulose. The macromolecular hydroxyl of cellulose was absorbed with water molecules in the form of hydrogen bonds, and more heat was needed to discharge water. If the final MC was too low, especially 0%, ICD system would easily catch fire.

The resistance to moisture removal when WFs are dried can be divided into internal resistance and external resistance [37]. The internal resistance is mainly related to the length–diameter ratio, MC, temperature, and other factors of the WFs. The external resistance is directly related to the airflow velocity affecting the mass exchange coefficient between the air and the WF surface. Airflow velocity was an important external factor affecting the drying of WFs. When the airflow velocity increased, the WFs had a short residence time in the equipment, and the MC showed an increasing trend. In contrast, when the inlet velocity increased from 11 to 13 m/s, the final MC showed a decreasing trend, indicating that the increased airflow velocity caused the boundary layer on the WF surface to be destroyed and caused the moisture on the WF surface to be quickly carried away as vapor. As heat and moisture transfer conditions were improved and the drying process was accelerated, properly increasing airflow rate was conducive to improving fiber drying efficiency.

Feed rate also has a certain influence on the final MC. As the feed rate increases continuously, the final MC shows an upward trend. When more fibers were invested simultaneously, the moisture in the pipe airflow was improved, which reduced the gradient of MC between the interior of the fiber and the surface, so that the rate of water migration was slower and the final MC was higher. Increasing the feed rate could reduce the drying efficiency because the unit energy consumption was unchanged, and the total energy consumption increased sharply [38]. Secondary drying or heating drying was required during the drying process. Therefore, if the feed rate was too high, the drying time would be too long, which could affect the next operation of ICD.

## Analysis of the ability of PSO-LSTM to predict MC

LSTM can deal with the gradient problem of neural networks in computation in information processing, maintain better accuracy, and solve the problem that traditional neural networks can only deal with the prediction of linear sequence. Twelve groups of sample data were selected as test sets. As shown in Fig 14, the LSTM had good fitting accuracy; its determination coefficient, $R^2$, is 0.9446, which is higher than that of RSM and BP (Table 6). Therefore, LSTM could be characterized as more stable. The LSTM model had the best prediction performance, followed by the BP method. Compared to the traditional BP, the LSTM could effectively learn the long-term dependence of process factors to achieve the ideal prediction effect. The relative

**Table 8. Comparison and verification of Different Methods with 96 groups test data for predicting MC.**

| Model | RMSE | MAE | MAPE |
|---|---|---|---|
| BP | 0.668 | 0.4743 | 0.9475 |
| LSTM | 0.515 | 0.2881 | 0.9552 |
| LSTM- PSO | 0.364 | 0.1282 | 0.9829 |

error value of the LSTM-PSO algorithm under the optimal process conditions was lower than that of the response surface and BP-PSO (Table 7), indicating that the MC prediction model based on the LSTM-PSO was effective in optimizing the process parameters of the ICD.

The expansion of the sample data to 400 groups was continued (Table 8), and the neural network model was constructed again by training. The results showed that the Pearson correlation coefficient of 96 groups of sample data is higher than that of 12 groups of test set data. The higher the resulting determination coefficient indicated that the expansion of the sample data capacity, the optimization effect of the prediction model was improved, indicating that this model has excellent predictive potential.

To verify the accuracy of the proposed scheme, BP, LSTM, and PSO-LSTM models were used to predict the MC on the same data set. By comparing 96 groups of data for the test set, it can be found that the results simulated by PSO-LSTM were better than those of LSTM model (Fig 16), the MC prediction value was closer to the actual value, and root MSE, MAE, and MAPE were better than other models (Table 8). In addition, the higher the resulting determination coefficient indicated that the expansion of the sample data capacity, the optimization effect of the prediction model was improved, indicating that this model has excellent predictive potential.

## Conclusion

The investigation into the energy consumption of ICD, oven-drying, and drum-drying of poplar fibers revealed that the ICD had lower energy consumption and lower costs. By comparing the MLR model, BP neural network model, and LSTM neural network model, the prediction model of MC in WFs dried by ICD could be built. The variance analysis of RSM showed that there was a highly significant relationship between the process factors of the regression model and the final MC of poplar fiber. The initial MC and inlet air temperature in the model were extremely significant ($P < 0.001$), while the feed rate was generally significant ($P < 0.05$). The MSE and MAPE of the LSTM model were smaller than those of MLR model and BP model. Under the PSO algorithm, the final MC of LSTM obtained by optimization was 0.96%, and the error was lower than 1.33% and 1.43% obtained by the RSM model and the BP model, respectively. The PSO-LSTM method was more suitable for the process optimization and the prediction of the MC of WF. The analysis method adopted in this study could provide a reference for optimizing the process of WF WPCs and lay a theoretical foundation for the application of ICD technology in the biomass composite industry.

In this study, only some process parameters were considered for the final MC. In future studies, several factors can be considered, such as the length–diameter ratio and wood species. In addition, the PSO-LSTM model was combined with other models to further verify the prediction accuracy of the model.

## Supporting information

**S1 File.**
(DOCX)

## Acknowledgments

Feng Chen conceived and designed the experiments; Xun Gao performed the experiments; Jing Xu and Xinghua Xia analyzed the data and wrote the paper. We thank Editage (www. editage.com) for editing manuscript to ensure language and grammar accuracy.

## Author Contributions

**Conceptualization:** Feng Chen.

**Data curation:** Xun Gao.

**Methodology:** Xun Gao.

**Software:** Jing Xu.

**Visualization:** Xinghua Xia.

**Writing – original draft:** Feng Chen.

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
