## [Decision Letter · Decision Letter 0]

22 Oct 2021

PONE-D-21-29751USING Long Short-Term Memory neural network and Particle Swarm Optimizer techniques for predicting moisture content of poplar fibers by Impulse-cyclone air drying systemPLOS ONE

Dear Dr. Chen,

Thank you for submitting your manuscript to PLOS ONE. After careful consideration, we feel that it has merit but does not fully meet PLOS ONE’s publication criteria as it currently stands. Therefore, we invite you to submit a revised version of the manuscript that addresses the points raised during the review process.

 Based on the comments received from the reviewers and my own observation, I recommend major revisions

We look forward to receiving your revised manuscript.

Kind regards,

Thippa Reddy Gadekallu

Academic Editor

PLOS ONE

Journal Requirements:

[The National Natural Science Foundation of China (Grant No.31901243) financially supported this research. Feng Chen conceived and designed the experiments; Xun Gao performed the experiments; Jing Xu and Xinghua Xia analyzed the data and wrote the paper.]

 [This research was funded by the National Natural Science Foundation of China (Grant No.31901243).]

Reviewers' comments:

Reviewer's Responses to Questions

**Comments to the Author**

1. Is the manuscript technically sound, and do the data support the conclusions?

Reviewer #1: Yes

Reviewer #2: Yes

2. Has the statistical analysis been performed appropriately and rigorously? 

Reviewer #1: No

Reviewer #2: I Don't Know

3. Have the authors made all data underlying the findings in their manuscript fully available?

Reviewer #1: No

Reviewer #2: Yes

4. Is the manuscript presented in an intelligible fashion and written in standard English?

Reviewer #1: No

Reviewer #2: Yes

5. Review Comments to the Author

Reviewer #1: Please consider the following comments to improve your manuscript before your resubmission:

1. The is no proper literature review on the existing algorithms applied on the same application.

2. There is no justification on why LSTM-NN, BP-NN, and RSM are considered to be compared.

3. There is no justification on why PSO is considered to optimize the model? Why not any other optimization technique?

4. There is no justification why 70/30 data split is considered and not any other split method.

5. No significant analysis. I suggest to have one.

6. Please proof read the manuscript before the resubmission.

Reviewer #2: In my opinion, the paper is well written and has good technical components, and is clearly described but a rewrite is required before acceptance. I have some suggestions and questions.

1. Abstract is unnecessarily wordy. Make it brief and concise. Also, Conclusion should clearly state the outcome. Some of the obtained results need to be highlighted in the conclusion section.‎

2. There are several number of techniques have been described in Introduction section. How do the authors outperform the each of these reviewed system? A clear statement is needed to highlight the contribution.

3. Methodology is not clear. Provide an algorithm and flowchart of the whole work. The authors need to add a new figure to show the main structure of the proposed system. ‎This will help the reader to get a better understanding of what is going on in the proposed ‎system.‎

4. There are lots of typos. English needs to revise again with a professional editing service. Also, the figures are not clear and in poor quality.

5. Mention the limitations of the developed system elaborately.

6. For any research paper to prove the validity of their system, the results must be compared with existing systems. This was lack in the paper. Try to compare the results with existing systems.

6. PLOS authors have the option to publish the peer review history of their article (what does this mean?). If published, this will include your full peer review and any attached files.

Reviewer #1: No

Reviewer #2: **Yes: **Dr. Kadiyala Ramana

---

## [Author Response · Author response to Decision Letter 0]

15 Feb 2022

Response to reviewers’ comments

Dear Managing Editor,

Thanks so much for your professional review of our manuscript entitled “ Using LSTM and PSO Techniques for Predicting Moisture Content of Poplar Fibers by Impulse-cyclone Drying ” (ID:PONE-D-21-29751). We also highly appreciate the beneficial suggestions and comments from the reviewers. All the questions pointed out by reviewers have been answered carefully and discussed in detail. The changes are marked with GREEN highlight font in the revised manuscript with tracked changes. The main corrections in the paper and the responses to the reviewer’s comments are listed as follows in the “Responses to the Reviewers”. We hope that these revisions are satisfactory and that the revised version is now suitable for publication in PLOS ONE. 

Thank you very much for your work concerning our article. 

Sincerely yours,

Feng Chen

Journal Requirements:

1.Please ensure that your manuscript meets PLOS ONE's style requirements, including those for file naming. The PLOS ONE style templates can be found at https://journals.plos.org/plosone/s/file?id=wjVg/PLOSOne_formatting_sample_main_body.pdf and https://journals.plos.org/plosone/s/file?id=ba62/PLOSOne_formatting_sample_title_authors_affiliations.pdf

Response: Thank you for this good suggestion. We have modified the format according to the requirements of the journal.

[The National Natural Science Foundation of China (Grant No.31901243) financially supported this research. Feng Chen conceived and designed the experiments; Xun Gao performed the experiments; Jing Xu and Xinghua Xia analyzed the data and wrote the paper.]

 [This research was funded by the National Natural Science Foundation of China (Grant No.31901243).]

Response: Thank you for this good suggestion. We have eliminated it and the changes can be found on Page 16 in the revised manuscript.

Review Comments to the Author：

Reviewer #1: 

1.The is no proper literature review on the existing algorithms applied on the same application

Response: Thank you for your congenial suggestion. We revised the literature review on line 48-120. And added the new references on line 603-716.

2. There is no justification on why LSTM-NN, BP-NN, and RSM are considered to be compared.

Response: Thank you for this good suggestion. Multiple linear regression model, such as RSM, is a traditional method to predict moisture content, and BP neutral network model is a common neural network. However, the current model is usually a shallow architecture with only one hidden layer to deal with non-linear problems. Limited power makes it impossible to make accurate judgment and MC prediction when facing complex process conditions, such as ICAD. Therefore, LSTM neural network was used in this paper to improve the accuracy of prediction model. We have focused on this issue on line 85-110. 

3. There is no justification on why PSO is considered to optimize the model? Why not any other optimization technique?

Response: Thank you for pointing this out. Particle swarm optimization (PSO) and genetic algorithm (GA) are both optimization algorithms, both of which belong to bionic algorithm. We optimized both algorithms in the experiment, but GA algorithm could not give stable results, so we chose PSO as the optimization method. In view of this problem, we will further compare the optimization effects of GA and PSO algorithms in the future.

4.There is no justification why 70/30 data split is considered and not any other split method.

Response: Thank you for pointing this out. In traditional machine learning, the 70/30 split principle is commonly used, that is, 70% of the entire data set is used for the training of the model. 30% of the entire data set is used for testing the model. The above data allocation is not mandatory, but provided by experience. Therefore, we adopted the 70/30 data split in the selection of data sets. 

5. No significant analysis. I suggest to have one.

Response: Thank you for your congenial suggestion. We added relevant experiments to demonstrate the rationality of the results, and carried out in-depth analysis in the results and discussion. 

6. Please proof read the manuscript before the resubmission.

Response: Thanks. We have read the article carefully and modified the text. English expression has been carefully improved throughout the manuscript. The touch up certificate is submitted as an attachment

Reviewer #2: 

1.Abstract is unnecessarily wordy. Make it brief and concise. Also, Conclusion should clearly state the outcome. Some of the obtained results need to be highlighted in the conclusion section.‎

Response: Thank you for pointing this out. Considering the Reviewer’s suggestion, we have revised abstract and the changes can be found on Page 2, line 31-46 in the revised manuscript.

2. There are several number of techniques have been described in Introduction section. How do the authors outperform the each of these reviewed system? A clear statement is needed to highlight the contribution.

Response: Thank you for this good suggestion. As for the ICAD drying technology proposed in this paper, we have done detailed preliminary research and published relevant papers, proving that this technology is feasible in the drying of wood fiber, and we have obtained relevant financial support for this technology. 

In terms of moisture content prediction model, we searched relevant papers and literature. At present, there are only two methods for MC prediction: multiple linear regression and machine learning, while multiple linear regression cannot be used for multivariate model prediction. There are many models in machine learning, such as BP neural network, LSTM neural network, CNN neural network, and compound neural network, etc. In the experimental model, we conducted the model prediction based on the commonly used RSM multiple linear regression, BP neural network and LSTM neural network and combined with the PSO optimization algorithm. The results showed that PSO-LSTM neural network has the best prediction effect. For other methods, such as CNN and GA, we will further analyze and compare them in the next paper.

3. Methodology is not clear. Provide an algorithm and flowchart of the whole work. The authors need to add a new figure to show the main structure of the proposed system. ‎This will help the reader to get a better understanding of what is going on in the proposed ‎system.‎

Response: Thank you for this good suggestion. We have added the main structure of MC prediction model in Figure 5.

4. There are lots of typos. English needs to revise again with a professional editing service. Also, the figures are not clear and in poor quality.

Response: We have read the article carefully and modified the text. English expression has been carefully improved throughout the manuscript. The touch up certificate is submitted as an attachment. We have modified the figures in the manuscript.

5. Mention the limitations of the developed system elaborately.

Response: Several relevant parameters of drying system proposed in this paper were selected, but the current parameters are only for poplar fiber, and the evaluation of other types of biomass fiber materials, such as straw and bamboo fiber, has not been analyzed. Therefore, the application of artificial neural network to poplar tree species does not represent the prediction of moisture content of all natural fiber materials. Therefore, in future research, tree species need to be analyzed as research variables. In addition, the drying process is a very complex process, and the uniformity of fiber will also affect the prediction results of moisture content. For fibers with various aspect ratios, the prediction of this model will also have some limitations.

6. For any research paper to prove the validity of their system, the results must be compared with existing systems. This was lack in the paper. Try to compare the results with existing systems.

Response: Thank you for your kind suggestion. We have reinterpreted the importance of wood fiber pretreatment in the production of WPC. The revision can be found on line Page 2, 64 to 79. Compared with other conventional drying technologies, ICAD has uniform fiber moisture content, fast speed and low energy consumption. We discussed the energy consumption and moisture content uniformity of this technology and conventional drying on Page 9, line 353 to 385. However, the operability of ICAD was often based on the operator's experience and lacked the drying benchmark of operation. It was necessary to study the drying characteristics of the ICAD system. Based on the prediction of moisture content, the drying process conditions can be determined in advance to improve the reliability and operability of the overall operation of the equipment. Therefore，in this paper, a water content prediction method based on LSTM-PSO neural network is proposed, which can obtain the final water content directly from the original data and then make an accurate prediction of ICAD drying process.

---

## [Decision Letter · Decision Letter 1]

16 Mar 2022

Using LSTM and PSO Techniques for Predicting Moisture Content of Poplar Fibers by Impulse-cyclone Drying

PONE-D-21-29751R1

Dear Dr. Chen,

We’re pleased to inform you that your manuscript has been judged scientifically suitable for publication and will be formally accepted for publication once it meets all outstanding technical requirements.

Kind regards,

Thippa Reddy Gadekallu

Academic Editor

PLOS ONE

Additional Editor Comments (optional):

Reviewers' comments:

Reviewer's Responses to Questions

**Comments to the Author**

1. If the authors have adequately addressed your comments raised in a previous round of review and you feel that this manuscript is now acceptable for publication, you may indicate that here to bypass the “Comments to the Author” section, enter your conflict of interest statement in the “Confidential to Editor” section, and submit your "Accept" recommendation.

Reviewer #2: All comments have been addressed

2. Is the manuscript technically sound, and do the data support the conclusions?

Reviewer #2: (No Response)

3. Has the statistical analysis been performed appropriately and rigorously? 

Reviewer #2: (No Response)

4. Have the authors made all data underlying the findings in their manuscript fully available?

Reviewer #2: (No Response)

5. Is the manuscript presented in an intelligible fashion and written in standard English?

Reviewer #2: (No Response)

6. Review Comments to the Author

Reviewer #2: (No Response)

7. PLOS authors have the option to publish the peer review history of their article (what does this mean?). If published, this will include your full peer review and any attached files.

Reviewer #2: **Yes: **Dr. Kadiyala Ramana

---

## [Editor Report · Acceptance letter]

29 Mar 2022

PONE-D-21-29751R1 

Using LSTM and PSO Techniques for Predicting Moisture Content of Poplar Fibers by Impulse-cyclone Drying 

Dear Dr. Chen:

I'm pleased to inform you that your manuscript has been deemed suitable for publication in PLOS ONE. Congratulations! Your manuscript is now with our production department. 

Kind regards, 

on behalf of

Dr. Thippa Reddy Gadekallu 

Academic Editor

PLOS ONE